

# Atmospheric observations and emission estimates of ozone-depleting chlorocarbons from India

Daniel Say[1], Anita L. Ganesan[2], Mark F. Lunt[3], Matthew Rigby[1], Simon O'Doherty[1], Chris Harth[4], Alistair J. Manning[5], Paul B. Krummel[6], and Stephane Bauguitte[7]

[1]School of Chemistry, University of Bristol, Bristol, BS8 1TS, UK
[2]School of Geographical Sciences, University of Bristol, Bristol BS8 1SS, UK
[3]School of Geosciences, University of Edinburgh, Edinburgh, EH9 3JW, UK
[4]Scripps Institution of Oceanography, University of California, San Diego, La Jolla, USA
[5]Met Office Hadley Centre, Exeter, EX1 3PB, UK
[6]Climate Science Centre, CSIRO Oceans and Atmosphere, Aspendale, Australia
[7]Facility for Airborne Atmospheric Measurements, Cranfield University, MK43 0AL, UK

**Correspondence:** Daniel Say (Dan.Say@bristol.ac.uk)

**Abstract.** While the Montreal Protocol has been successful in reducing emissions of many long-lived ozone-depleting substances, growth in the global emission rates of unregulated very short-lived substances (VSLS) poses a potential threat to the recovery of the ozone layer. The sources of these VSLS are not well-constrained, with major source regions poorly monitored by existing measurement networks. Given India's rapidly growing economy, its emissions of both regulated chlorocarbons and unregulated VSLS chlorocarbons are suspected to have global significance. Furthermore, VSLS from the Asian monsoon regions have a greater impact on ozone-depletion than those emitted elsewhere due to the ability of monsoon systems to rapidly transport pollutants to the lower stratosphere. Previous atmospheric measurements of chlorocarbons from the Indian sub-continent are scarce. Here we present a new set of observations, derived from flask samples collected during a 2-month aircraft campaign in India and use these measurements to infer India's chlorocarbon emissions. We show that emissions of carbon tetrachloride from northern and central India (2.3 (1.5 – 3.4) Gg yr$^{-1}$), are likely due to inadvertent production and release during the manufacture of chloromethanes (specifically dichloromethane and chloroform) and account for approximately 7% of the global total. Emissions of methyl chloroform from the same region were estimated to be 0.07 (0.04 - 0.10) Gg yr$^{-1}$ which account for less than 5% of remaining global emissions. We used a population scaling to estimate India's emissions of the very short-lived chlorocarbons dichloromethane, perchloroethene and chloroform, which were estimated to be 69.2 (55.8 - 82.9) Gg yr$^{-1}$, 2.9 (2.5 – 3.3) Gg yr$^{-1}$ and 25.7 (21.6 – 29.9) Gg yr$^{-1}$ respectively. In the case of dichloromethane, our estimate is consistent with a 3-fold increase in emissions since the last estimate derived from atmospheric data in 2008.

*Copyright statement.* TEXT



# 1 Introduction

Chlorinated methanes, ethanes and ethenes (herein referred to as chlorocarbons) are emitted to the atmosphere from their widespread use as solvents and as feedstocks in the manufacture of refrigerants including hydrochlorofluorocarbons (HCFCs) and hydrofluorocarbons (HFCs). Two of these gases, carbon tetrachloride ($CCl_4$, herein referred to as CTC) and methyl chloroform ($CH_3CCl_3$, herein referred to as MCF), were recognised as major contributors to stratospheric ozone depletion; hence, their production and consumption is regulated under the Montreal Protocol on Substances that Deplete the Ozone Layer.

Originally used as a cleaning agent, the production of CTC for dispersive applications was banned globally in 2010. While 'bottom-up' emissions of CTC, estimated from reported consumption for feedstock use, are small ($1 - 4$ Gg yr$^{-1}$ (Montzka et al., 2011)), top-down studies, based on atmospheric observations, suggest actual global emissions still exceed 30 Gg yr$^{-1}$ (Lunt et al., 2018; Liang et al., 2014; Chipperfield et al., 2016). In a recent bottom-up study, Sherry et al. (2018) estimated that much of these additional emissions occur during the production of chloromethanes and perchloroethene and in chlor-alkali facilities. In contrast to CTC, global emissions of MCF, used for cold cleaning and as a degreaser for precision engineered components, have declined dramatically as a result of the Montreal Protocol, although there is evidence for continuing low-level emissions from some regions (e.g. Maione et al. (2014)).

Because of their comparatively short lifetimes, the chlorocarbons dichloromethane ($CH_2Cl_2$, herein referred to as DCM), perchloroethene ($C_2Cl_4$, herein referred to as PCE) and chloroform ($CHCl_3$) (all of which are classified as very short-lived substances (VSLS)), were not regulated under the Montreal Protocol. While the ozone-depletion potentials (ODP) of these gases are considerably smaller than CTC or MCF (Table 1), global emissions of DCM and chloroform are much larger (Carpenter et al., 2014). In addition, emissions of DCM have increased substantially in recent years. Hossaini et al. (2017) estimated global emissions to be ~0.6 Tg yr$^{-1}$ in 2004, which rose to over 1.1 Tg yr$^{-1}$ by 2014. If this trend continues, Hossaini et al. (2017) shows that DCM emissions could lead to an appreciable delay in the recovery of the Antarctic ozone-hole by 17 - 30 years. Likewise, Fang et al. (2018) (in press) estimates that continued growth in global emissions of chloroform could result in a further delay in ozone layer recovery of $6 - 11$ years. The study highlighted eastern China as a major contributor to global emissions.

The global budgets of DCM and PCE are dominated by anthropogenic sources, most notably from their use as cleaning agents and degreasers. PCE is also produced in small quantities as a result of petroleum and coal combustion (McCulloch et al., 1999). In contrast, global emissions of chloroform have both anthropogenic and biogenic sources. Historically, 90% of chloroform present in the atmosphere has been attributed to biogenic sources. However, more recent studies have suggested that anthropogenic sources may have been significantly under-estimated. Trudinger et al. (2004) estimated that these anthropogenic sources, which include fluoropolymer production, paper manufacture and water chlorination, could account for as much as 50% of the global budget.

In certain regions of the world, large-scale convective systems provide an efficient route for the transport of VSLS to the stratosphere before they are substantially oxidised in the troposphere. South Asia's monsoon systems, similar to those over eastern Asia, provide one such pathway (Fadnavis et al., 2013; Randel et al., 2010). This is particularly important for the



halogenated VSLS, which are unregulated primarily because they are expected to be destroyed quickly in the troposphere. Brioude et al. (2010) show that VSLS halocarbons emitted from South Asia have ODPs up to 8 times greater than those emitted from elsewhere in Asia, and 22 times greater than emissions from Europe.

India is the world's second most populous country and seventh largest economy, yet little is known about its emissions of chlorocarbons. Previous measurements from South Asia were either made at a considerable distance from India's source regions, or at high altitude. Maione et al. (2011) described three years of halocarbon measurements derived from flask samples collected at the Nepal Climate Observatory – Pyramid (NCO-P), located in the Khumbu valley at 5079 m asl. The measurements suggested non-zero MCF emissions from the Indian subcontinent in 2008. Using flask samples collected during the CARIBIC programme (Civil Aircraft for Regular Investigation of the Atmosphere Based on an Instrument Container (Brenninkmeijer et al., 2007)), Leedham Elvidge et al. (2015) characterised DCM emissions from the Indian sub-continent. Whole air samples were collected at altitudes between 10 – 12 km at locations characteristic of South Asian monsoon outflow. Using carbon monoxide as a tracer, they estimated Indian DCM emissions of 4.9 (2.7 – 7.2) Gg yr$^{-1}$ in 1998, rising to 20.3 (15.8 – 24.8) Gg yr$^{-1}$ in 2008. Oram et al. (2017) also reported flask measurements from CARIBIC, characterising South Asian emissions of DCM and PCE in 2012 - 2014. While many of these samples were collected directly over northern India, most were collected at altitudes of 10 – 12 km and were found to be more sensitive to South East Asia than the Indian sub-continent.

To the best of our knowledge, India does not report national emissions estimates for any of the chlorocarbons discussed in this study. Until 2010, when non-Article 5 countries were required to cease the production and consumption of CTC and MCF for dispersive use, India did report its consumption to the United Nations Environment Program (UNEP). In 2010, India reported consumption of 0 Gg yr$^{-1}$ for both gases. Bottom-up estimates were made for a number of the unregulated chlorocarbons as part of the Reactive Chlorine Emissions Inventory. McCulloch et al. (1999) shows India's DCM and PCE emissions to be 11.1 Gg yr$^{-1}$ and 6.0 Gg yr$^{-1}$ respectively, in 1990. However, since these estimates were made more than two decades ago, they are unlikely to be representative of more recent emissions.

In this study, we present a new set of atmospheric measurements, derived from flask samples collected during a 2-month aircraft campaign. Over 90% of these samples were collected below 1.5 km altitude, providing high sensitivity to surface emissions from India compared to previous measurements. We use these measurements to estimate India's chlorocarbon emissions, providing a new benchmark against which future growth in emissions from the sub-continent may be evaluated.

## 2 Materials and Methods

### 2.1 Sample collection and analysis

176 flask samples were collected on board the FAAM (Facility for Airborne Atmospheric Measurements) BAe-146 research aircraft between $12^{th}$ June and the $9^{th}$ July 2016 (Table 2). Ambient air was drawn into the sample lines from a forward facing inlet on the exterior of the aircraft using a metal bellows pump (Senior Aerospace PWSC 28823-7) and compressed to a maximum of 41 psig in evacuated 3 L stainless steel canisters (SilcoCan, Restek), giving a maximum usable volume of 9 L. Each sample case (including flasks and sample lines) was evacuated pre-flight to a maximum pressure of $1 \times 10^{-5}$





psig, and a maximum of 64 flasks were available to fill on each flight. Sample fill duration varied by altitude and ranged from 25 - 60 seconds, roughly equivalent to 7 km of flight track at typical cruising velocity. Flasks were collected at regular intervals throughout each flight, and at times of polluted air mass detection as indicated by an in situ carbon monoxide analyser (Aerolaser AL5002). When not in use, sample cases were stored in a shipping container and with the exception of flasks

collected over the Arabian Sea, returned to the University of Bristol for analysis within a month of collection. Flask sample locations are shown in Fig. 1.

Chlorocarbon detection was achieved via the Medusa Gas Chromatography Spectrometry (Medusa GCMS) analytical system (Miller et al., 2008; Arnold et al., 2012). Each flask was analysed three times in total. Sample volume (1.75 L) and injection flow rate (50 cm$^3$ min$^{-1}$) were reduced compared to those described by Miller et al. (2008), in order to account for the reduced

pressure of the flask samples. For each gas, mole fractions are reported relative to those of a working standard, analyses of which bracketed the flask runs. The reference gas concentrations are linked to a suite of gravimetrically prepared 'primary' standards via a hierarchy of compressed gas cylinders. A detailed description of the calibration routine can be found in Miller et al. (2008). Bi-monthly system blanks showed no interference from potential laboratory sources/carrier gas impurities for the chlorocarbons discussed here. For each gas, routine monitoring of the ratio of target to qualifier ion(s) ensured that there was

no interference from co-eluting species. Mean measurement precisions, estimated as the standard deviation of the three repeat flask analyses, were estimated at 1.3%, 1.7%, 1%, 1%, 0.5%, for CTC, MCF, DCM, PCE and chloroform respectively.

## 2.2 Numerical Atmospheric Modelling Environment (NAME)

The Met Office Lagrangian particle dispersion model, NAME (Numerical Atmospheric dispersion Modelling Environment), was run in backwards mode (Manning et al., 2011) to simulate 30-day back-trajectories for each minute along each flight path.

Simulations that corresponded with sample collection were selected to quantify the influence of surface (defined as 0 - 40 m above ground level) fluxes on each atmospheric measurement. NAME was driven by meteorological fields extracted from the operational analysis of the UK Met Office Unified Model (UM), at an approximate horizontal resolution of 17 km in 2016. The model domain spanned from 55 – 109 °E and 6 – 48 °N and to a maximum altitude of 19 km. For each simulation, particles were released from a cuboid whose dimensions were defined as the change in latitude, longitude and altitude of the aircraft

during that one minute period, at a rate of 1000 particles min$^{-1}$. Wherever possible, samples were collected during periods of level flight, to reduce the area of the particle release cuboid and hence minimise potential transport errors. The ability of NAME to accurately simulate meteorological parameters (wind speed and direction) is discussed in our companion paper, Say et al. (2019). In general, good agreement was found between measured and modelled meteorological parameters, and where small discrepancies were observed, they were shown to have no significant effect on the posterior emission estimates. The time

and 3-dimensional point at which each particle exited the model domain was recorded to provide sensitivity to the boundary conditions.

Given the short lifetimes of DCM, chloroform and PCE, there is the potential for chemical loss during a typical 30-day simulation. Fang et al. (2018) investigated the impact of modelling short-lived substances with lifetimes of around six months over regional domains, without accounting for loss processes. Their study showed that, for sources that are within several





hundred kilometres of measurement locations, as in this set-up, the decay is very small (less than 1%) over the time-scales of transport from source to receptor and can thus be neglected.

## 2.3 Estimating emissions using a hierarchical trans-dimensional inverse modelling technique

The hierarchical trans-dimensional Bayesian framework employed in this study has been applied to derive halocarbon emissions from elsewhere in Asia (Lunt et al., 2018). The underlying principles of reversible jump Markov Chain Monte Carlo (rj-MCMC) and the way in which this method derives the resolution of the inversion are described in Lunt et al. (2016). The hierarchical treatment of uncertainties and the principals of hierarchical Bayesian modelling are described by Ganesan et al. (2014). For each chlorocarbon, the rj-MCMC algorithm was ran 400,000 times, with the first 100,000 of these iterations discarded as 'burn-in' to ensure the system had no knowledge of the initial state. The remaining 300,000 iterations were then thinned by storing every $100^{th}$ iteration, with the posterior PDFs formed from the resulting 3000 samples. In the following sections, reported emissions estimates refer to the mean of our posterior PDFs, with uncertainties formed from the $5^{th}$ and $95^{th}$ percentiles of the PDF.

Our derived emission fields were aggregated over a region that corresponded approximately to the maximum sensitivity of the atmospheric measurements. This region was denoted northern-central India (herein referred to as NCI, Fig. 1). We did not aggregate posterior emissions over all of India because emissions from Southern and far North East India would mainly reflect the prior due to the low sensitivity in the measurements. We did not attempt to estimate a national total for CTC and MCF, since the suspected remaining sources of these gases (such as fugitive industrial emissions) are not necessarily linked to population. However, for DCM, PCE and the anthropogenic sources of chloroform, previous studies have shown a strong link between population and emissions (Aucott et al., 1999; McCulloch et al., 1999). Hence, for these gases we estimated emissions for the whole of India using population statistics (CIESEN, 2016) – the NCI model domain was calculated to account for 71.7% of India's population in 2016.

The emission estimates discussed below are based on measurements collected over a two-month period. For gases whose emissions do not vary by season, these estimates are likely to be consistent with annual emissions. While significant seasonality has been reported for anthropogenic halocarbons such as the refrigerant HCFC-22 (Xiang et al., 2014), chlorocarbon emissions arising from industrial processes and landfill are unlikely to have large seasonal cycles. Biogenic sources of chloroform have been shown to exhibit significant seasonality (Laturnus et al., 2002), yet emissions from anthropogenic activities (e.g. use as a feedstock) are not likely to vary by season. Due to the nature of aircraft sampling, our national, annual estimates are likely to be representative for gases whose emissions are both widespread and do not exhibit significant variation in time. These criteria are thought to be met for all the chlorocarbons studied here.

## 2.4 A priori emissions

Little detailed information is available on India's chlorocarbon emissions. India's CTC emissions were estimated at 2.8 Gg yr$^{-1}$ based on the 2014 estimate by Sherry et al. (2018). A priori MCF emissions were estimated using a population-based scaling of the global total derived using the AGAGE 12-box model (Rigby et al., 2014). India was assumed to account for 17.7% of the



global population in 2016 (CIESEN, 2016), giving an emissions estimate of 0.3 Gg yr$^{-1}$. For DCM, a priori emissions were taken from the only available top-down study conducted in the region - Leedham Elvidge et al. (2015) estimated India's DCM emissions to be 20.3 Gg yr$^{-1}$ in 2012. Terrestrial chloroform emissions were taken from the AGAGE 12-box model. Using a population scaling for India and assuming that 45% of chloroform emissions (biogenic and anthropogenic) originate on land

(McCulloch, 2003), India's land-based chloroform emissions were estimated at 3.0 Gg yr$^{-1}$. Oceanic chloroform emissions were adapted from Khalil et al. (1999), who estimated a northern hemispheric tropical ocean source of 50 Gg yr$^{-1}$. The ocean within our model domain was estimated to account for 8.3% of this source by area, equivalent to 4.2 Gg yr$^{-1}$. India's PCE emissions were taken from the Reactive Chlorine Emissions Inventory (McCulloch et al., 1999) and were estimated at 6.0 Gg yr$^{-1}$.

Information regarding the spatial distribution of these emissions is scarce, though India's emissions of DCM and PCE have previously been linked to population density (McCulloch et al., 1999). For chlorocarbons that are assumed to originate predominantly as a result of anthropogenic activities (CTC, MCF, DCM and PCE), prior emissions were distributed across the model domain using night light data provided by the National Oceanic and Atmospheric Administration (NOAA) DMSP-OLR satellite, which is made available at 30 arc second increments (https://ngdc.noaa.gov/eog/data/web_data/v4composites/). The

night-lights distribution was used such that areas of greatest light intensity correspond to the greatest a priori emissions. Night light data have been shown to correlate well with population density (Raupach et al., 2010), but have the advantage of also incorporating industrial sites such as chloromethane and PCE manufacturing facilities. For chloroform, both terrestrial and oceanic emission totals were distributed uniformly across the domain and this allowed the inversion to adjust emissions from both anthropogenic and biogenic source regions. For all species, the prior uncertainty of each spatial basis function was set to

100% of the prior mean but was described by a uniform PDF with lower and upper bounds of 50% and 500% respectively – the large uncertainty reflecting the lack of detailed information currently available for India. Large prior uncertainties mean that our posterior emissions over the NCI are informed almost entirely by the atmospheric measurements. To confirm that our posterior estimates were independent of the spatial distribution of the prior, results derived using the night-lights data were compared to those derived from a spatially uniform prior (see Results).

**2.5   A priori boundary conditions**

The back-trajectories simulated by NAME only consider emissions from within the model domain. In the absence of a reliable global spatial emissions inventory for these gases, a priori boundary conditions were estimated from the AGAGE 12-box model (Rigby et al., 2014). Uniform 'curtains' were assigned to the North, East, South and West boundaries of the NAME model domain, and for each curtain, a mole fraction was assigned from a semi-hemisphere resolved by the 12-box model.

The a priori mole fraction for the North boundary was based on the box model estimate for 30 – 90 °N, the East and West boundaries from 0 – 30 °N, and the South boundary from 0 – 30 °S. Offsets to the mole fraction curtains in each direction were solved for as additional parameters in the inversion.



## 2.6 Global emissions estimates using the AGAGE 12-box model

Chlorocarbon emissions from the NCI and India where appropriate are compared to global emissions estimated using the AGAGE 12-box model and baseline atmospheric data from five remote AGAGE stations (Mace Head, Ireland; Trinidad Head, USA; Ragged Point, Barbados; Cape Matatula, American Samoa and Cape Grim, Tasmania (Prinn et al., 2018)). The model
uses annually repeating meteorology and a hydroxyl field climatology described by Spivakovsky et al. (2000), tuned to match the atmospheric growth rate of MCF. Modelled atmospheric lifetimes, derived using the temperature-dependent rate constants for the reaction of each chlorocarbon with OH (Burkholder et al., 2015), were estimated at 32.9, 5.0, 0.5, 0.4 and 0.6 years for CTC, MCF, DCM, PCE and chloroform respectively. The inversion propagates uncertainties in the observations, including instrumental precision and potential errors in the calibration scale of each gas, in addition to uncertainty in the assumed at-
mospheric lifetime, to the posterior solution. For DCM, PCE and chloroform, previous work has shown that the box model's inability to resolve longitudinal gradients and/or its coarse latitudinal and vertical resolution can lead to substantial differences in derived emissions when different monitoring networks are used (Engel and Rigby, 2019) (in press). Therefore, the uncertainties prescribed to the global emissions estimates presented here may be under-estimated. However, a thorough investigation of global uncertainties is beyond the scope of this study. A complete description of the AGAGE 12-box model inversion
framework is given in Rigby et al. (2014).

## 3   Results

### 3.1   Atmospheric measurements

Mole fractions of each chlorocarbon measured in the flask samples are shown in Fig. 2 in comparison to statistical baselines representative of northern and southern hemispheres. These baselines are based on observations from Mace Head (Ireland)
and Cape Grim (Tasmania) research stations, respectively. Because of South Asia's monsoon circulation patterns during June-August, the air over India typically originates from southerly latitudes and background mole fractions of trace gases measured during this period are typically more representative of the Southern Hemisphere.

Regional inverse models use enhancements in mole fraction above the regional background to estimate emissions. A large number of significant enhancements (defined herein with respect to the southern hemispheric baseline) were observed in all
chlorocarbons except for MCF. For each gas, variability in the mole fraction of samples collected over the Arabian Sea was significantly lower than for those samples collected over Northern and Central India, indicating that most enhancements were the result of terrestrial sources, as opposed to possible oceanic sources, for example, for chloroform or CTC.

For the regulated gases CTC and MCF, a small number of enhancements were observed. However, these were far less numerous than those of the short-lived chlorocarbons, suggesting emissions are less widespread. Maximum enhancements of
12 ppt and 26 ppt were observed for CTC and MCF respectively, though for MCF just four measurements were deemed to be enhanced. The sample exhibiting the highest MCF mole fraction was also found to have elevated concentrations of all the unregulated chlorocarbons discussed here, and the corresponding air-history indicated a possible source located in or near to





the coastal city of Visakhapatnam. In a recent study, Sherry et al. (2018) estimated that India's chloromethane manufacturing facilities may have produced as much as 20 Gg of CTC in 2014, though much of that is likely to have been destroyed, recycled or sold. India does not have any major operational facilities for the manufacture of PCE, which is another potential anthropogenic source of CTC (Sherry et al., 2018). In contrast, the remaining sources of MCF are harder to define. Reimann et al. (2005)

proposed that factories producing HCFC-141b and HCFC-142b were possible sources of MCF in Europe, since MCF is used as a feedstock in the production of these refrigerants. However, India only reports production of HCFC-22, the main feedstock of which is chloroform, while demand for other HCFCs is met entirely through imports (MoEFCC, 2017). Landfills are another possible source of MCF, with previous studies from other regions reporting emissions from municipal waste disposal facilities (Maione et al., 2014; Talaiekhozani et al., 2018).

Very large enhancements were found for DCM, with a maximum mole fraction of 1133 ppt (corresponding enhancement of 1120 ppt, Fig. 2). Samples collected at longitudes east of 81 °E were particularly enhanced above the baseline, suggesting that the flask samples were sensitive to regions producing/consuming large quantities of DCM as a solvent, feedstock or both. All three unregulated chlorocarbons exhibited significant elevations in mole fraction for samples collected around 77 °E, with maximum enhancements of 242 ppt, 69 ppt and 13 ppt for DCM, chloroform and PCE respectively. We found a significant (R

= 0.64) correlation between DCM and chloroform (Fig. 3), suggesting that these gases share similar sources. Since DCM is predominantly anthropogenic in origin, this correlation indicates that the majority of enhancements observed for chloroform are from anthropogenic sources, as opposed to natural origins. Both are used as feedstock gases for halocarbon manufacture (primarily for HFC-32 and HCFC-22, respectively). We previously reported a strong correlation (R = 0.58) between enhanced mole fractions of HFC-32 and DCM (Say et al., 2019), which suggests a strong linkage between the locations of DCM and

HFC-32 emissions.

A significant positive correlation (R = 0.69) was also found between PCE and DCM, gases for which anthropogenic sources dominate global emissions. For both gases, the largest enhancements were observed for samples collected downwind of New Delhi. PCE is used extensively in India as a dry cleaning agent (Srivastava, 2010). While biomass burning is a potential source of all three non-regulated chlorocarbons (Lobert et al., 1999; Rudolph et al., 1995), such activity is expected to be reduced

considerably during the wet monsoon season when our samples were taken.

### 3.2   Chlorocarbon emissions for NCI and India

Mean NCI and Indian emissions estimates and the relative contributions of each gas to 2016 global emissions are shown in Fig. 4 and Table 3. A comparison of the atmospheric measurements with the modelled mole fractions are given in Fig. 5. Emissions are also provided in ODP Gg for gases that have well-defined ODPs, which do not vary by region (CTC and MCF). ODP values

were taken from Carpenter et al. (2014) (Table 1). Tests were conducted to assess the sensitivity of our inversion to changes in the prior (night-light distributed versus spatially uniform) and potential transport model errors, e.g. the ability of NAME to accurately simulate atmospheric motion (Table 4). The latter is discussed in more detail in our companion paper, Say et al. (2019). In both instances and for all chlorocarbons, the results of these sensitivity tests were within the uncertainties of our posterior estimates.





### 3.2.1 Carbon tetrachloride

We estimate NCI CTC emissions to be 2.3 (1.5 - 3.4) Gg yr$^{-1}$ (1.7 (1.1 – 2.4) ODP Gg yr$^{-1}$), which accounts for 6.8 (4.4 - 10.0) % of global emissions in 2016. India reported that its production and consumption of CTC had ceased prior to 2016 (http://ozone.unep.org/countries/data). Since then, ongoing CTC emissions from India are not necessarily linked to its use as solvent but may persist due to fugitive leaks during chloromethane manufacture (most notably DCM and chloroform production) and from chlor-alkali plants. Sherry et al. (2018) estimated that India's chloromethane manufacturers might have produced as much as 20 Gg of CTC as by-product in 2014, with corresponding fugitive emissions of 2.8 Gg yr$^{-1}$, though these findings are not reflected in the UNEP reports. As these activities are not thought to be distributed evenly with respect to population, we do not scale NCI estimates to a national total. Our posterior emissions map (Fig. 6) is consistent with emissions from the chloromethane manufacturing facilities, while the known locations of chlor-alkali plants do not appear to be associated with large emissions. India's CTC emissions remain small compared to those of eastern China, whose average emissions from 2011 - 2015 were estimated at 17 (11 – 24) Gg yr$^{-1}$ (Lunt et al., 2018), but are of similar magnitude to those of the US, estimated at 4.0 (2.0 – 6.5) Gg yr$^{-1}$ between 2008 – 2012 (Hu et al., 2016). However, the majority of US emissions were attributed to chlor-alkali production plants, which differs from our finding that CTC emissions in India do not correspond with known locations of chlor-alkali production.

Carbon tetrachloride is typically produced as a by-product of chloroform manufacture at an estimated rate of 3-5% (Oram et al., 2017; Sherry et al., 2018) and where the production rate of chloroform is known, the total volume of CTC produced may also be inferred. While we were unable to find any chloroform production data for India, as noted above, chloroform is used as a feedstock in the production of the refrigerant HCFC-22. Over 99% of chloroform produced globally is used in the manufacture of HCFC-22, with 1 kg of HCFC-22 requiring 1.5 kg of chloroform as feedstock (Oram et al., 2017). Based on an extrapolation of reported HCFC-22 production statistics in India (available from 2006 – 2015 (UNEP, 2017)), we estimate India's HCFC-22 production in 2016 to be 55 Gg. If all chloroform produced was used for HCFC-22 manufacture, and all demand was met domestically (available data suggests India only imported ~165 tonnes of chloroform in 2016 (https://www.seair.co.in/chloroform-import-data.aspx), although we were unable to verify the completeness of this record), we estimate that India would produce 82.5 Gg of chloroform in 2016 and therefore, 2.5 – 4.1 Gg of CTC. As the majority of known CTC sources reside within the NCI, Indian emissions calculated using this 'bottom-up' method are consistent with our top-down estimate for NCI and the bottom-up study by Sherry et al. (2018).

### 3.2.2 Methyl chloroform

Based on its reports to the United Nations Environment Program (UNEP), India has not produced or consumed MCF since 2001. However, a small number of enhancements in the mole fraction of this gas strongly suggest that sources persist. As with CTC, the nature, location and magnitude of the sources of MCF are highly uncertain. Therefore, we do not attempt to estimate a total for the whole of India. At 0.07 (0.04 - 0.10) Gg yr$^{-1}$ (0.010 (0.006 – 0.014) ODP Gg yr$^{-1}$), MCF emissions from the NCI account for 4.1 (2.4 – 5.9) % of global emissions. Despite its status as a developing country, which meant India



had significantly longer to phase-out consumption of MCF when compared to developed countries, emissions from the NCI (which comprises 71.7% of India's population and includes several key industrial regions) are smaller than those from Europe, which were estimated to be 0.20 Gg yr$^{-1}$ in 2012 (Maione et al., 2014). Given the continued role of MCF in estimating global hydroxyl concentrations (e.g. Rigby et al. (2017)), further long-term measurements from India are required to better understand

the remaining sources of this gas, which may include release from landfill or fugitive leaks from halogen chemical plants.

### 3.2.3 Dichloromethane

We estimate Indian DCM emissions to be 69.2 (55.8 - 82.9) Gg yr$^{-1}$, and these contribute 7.6 (6.2 - 9.1) % of global emissions. When compared to previous estimates of India's DCM emissions, our results reflect substantial growth. Leedham Elvidge et al. (2015) estimated emissions of 4.9 (2.7 - 7.2) Gg yr$^{-1}$ in 1998, rising to 20.3 (15.8 - 24.8) Gg yr$^{-1}$ in 2008, suggesting a 2-

10 to 4-fold increase in emissions over that period. Our estimate is consistent with a similar growth trajectory. Our mean estimate represents an approximate 3-fold increase in emissions between 2008 and 2016. Global emissions over the same period rose from 611.5 Gg yr$^{-1}$ to 907.3 Gg yr$^{-1}$, representing an increase of 295.8 Gg yr$^{-1}$. The growth in India's emissions over this period (48.9 Gg yr$^{-1}$) would therefore represent 16.5% of the global rise. While estimates of Indian DCM emissions may have been increasing, they remain small when compared to those from China; Oram et al. (2017) estimated Chinese DCM emissions

to be 455 ± 45.5 Gg yr$^{-1}$ in 2015.

The rise in India's DCM emissions could be in part attributed to increased production of HFC-32, a widely used refrigerant. We show in our companion paper, Say et al., 2018, evidence for a lack of domestic consumption and propose that much of the HFC-32 produced in India is exported. It is possible that domestic demand for HFC-32 will increase in line with projected growth in sales of stationary air-conditioning units (Purohit et al., 2016). Production of HFC-32 in India is known to occur in

at-least one manufacturing facility in Rajasthan.

### 3.2.4 Perchloroethene

Emissions of PCE are almost exclusively anthropogenic in origin, due to its widespread use as a chemical intermediate and general-purpose solvent. Despite classification as a hazardous air pollutant by the United States Environmental Protection Agency (EPA, 2012), PCE is used extensively in India as a dry-cleaning solvent (Srivastava, 2010). We estimate India's PCE

emissions to be 2.9 (2.5 - 3.3) Gg yr$^{-1}$, which account for 3.5 (3.0 - 4.1) % of the global total. When compared to the only previous estimate of India's PCE emissions, which was calculated using bottom-up methods (3.9 Gg yr$^{-1}$ in 1990 (McCulloch et al., 1999)), our estimate either shows a discrepancy with bottom-up inventories or implies a decrease in emissions since 1990. The latter would be consistent with global emissions derived using the AGAGE 12-box model (Rigby et al., 2014), which have also declined steadily over the past 10 years, down from 124.2 (50.3 – 204.2) Gg yr$^{-1}$ in 2006 to 82.6 (35.8 – 133.3) Gg yr$^{-1}$

in 2016.



### 3.2.5 Chloroform

We estimate India's chloroform emissions to be 31.5 (28.7 – 34.3) Gg yr$^{-1}$, likely from anthropogenic sources due to the strong correlation of enhancements with DCM. These emissions account for 9.4 (8.6 – 10.2) % of global emissions in 2016. However, given the large biogenic component of global emissions, India's contribution to global anthropogenic emissions may

be significantly larger. Trudinger et al. (2004) estimated the anthropogenic component accounted for ~58% of global emissions in the year 2000. If we assume that the ratio of anthropogenic to biogenic sources has not changed significantly since that study, India could have accounted for ~16% of global anthropogenic chloroform emissions in 2016.

Production of HCFC-22 is the most common end-use of chloroform, accounting for around 99% of its consumption. India is the world's second largest producer of HCFC-22 after China (UNEP, 2017). Fang et al. (2018) (in press) estimate East

China's 2015 chloroform emissions to be 88 (80 - 95) Gg yr$^{-1}$, which are predominantly attributed to anthropogenic sources. Our findings are consistent with the work of Trudinger et al. (2004), which concluded that anthropogenic sources contribute a significantly greater proportion of the global chloroform budget than previously estimated (McCulloch, 2003).

### 4 Conclusions

The production and consumption for dispersive use of two chlorocarbons, CTC and MCF, is now banned under the Montreal

Protocol, but growth in emissions of unregulated very short-lived chlorocarbons pose a growing threat to ozone layer recovery. The current state of knowledge in India is restricted to historic bottom-up estimates of DCM and PCE (1990), a more recent (2014) bottom-up estimate for CTC, and a single top-down study (2008) for DCM. Whilst previous studies have collected atmospheric measurements that were representative of Indian chlorocarbon emission, these observations were confined to the lower stratosphere and hence have lower sensitivity to surface source regions than the samples presented in this work.

This study describes a new set of chlorocarbon measurements, collected from a research aircraft sampling mainly within the planetary boundary layer, below 1.5 km altitude, which exhibit strong sensitivity to surface emissions from northern and central India. We observed a small number of mole fraction enhancements for CTC and MCF, consistent with small ongoing emissions of these gases from India. The limited number of enhancements suggests that emissions are of these gases were not widespread in 2016. In contrast, a large number of significant enhancements were observed for all three unregulated very short-lived

chlorocarbons, indicative of strong sources across northern and central India. We found a strong correlation between DCM and chloroform, strongly suggesting that the majority of the enhancements found for chloroform were the result of anthropogenic activities, as opposed the biogenic sources.

India's chlorocarbon emissions were estimated using an inverse modelling framework. We estimate northern and central India's CTC and MCF emissions to be 2.3 (1.5 - 3.4) Gg yr$^{-1}$ accounting for 6.8% of global emissions, and 0.07 (0.04 - 0.10)

30 Gg yr$^{-1}$ accounting for 4.1 (2.4 – 5.9) % of global emissions, respectively. The results of our CTC inversion are consistent with unreported fugitive emissions arising from chloromethane (DCM and chloroform) manufacture. A population scaling was used to estimate India's emissions of the very short-lived chlorocarbons. At 69.2 (55.8 - 82.9) Gg yr$^{-1}$, India's 2016 DCM emissions are consistent with a 3- to 4-fold increase since 2008, with India responsible for 16.5% of the increase in global





emissions during that time. Our 2016 estimates provide a useful benchmark against which future changes to India's emissions of these gases can be assessed.

*Author contributions.* D.S. filled the sample flasks, conducted the analysis of samples, ran the inverse model and wrote the paper. A.G. coordinated the whole air sampling startegy, co-developed the inverse model code and contributed significantly to the writing of the paper.

5 M.L. co-developed the inverse model code. M.R. aided the interpretation of the inverse model output. S. O'D. co-conducted the instrumental analyses and is the station manager at Mace Head. C.H. produced the SIO chlorocarbon calibration scales. A.M. co-ran the NAME model. P.K. provided measurements from the Cape Grim atmospheric observatory. S.B. co-filled the sample flasks.

*Competing interests.* The authors declare that they have no conflict of interest.

*Acknowledgements.* The authors thank the personnel at the NERC Facility for Airborne Aircraft Measurements (FAAM), Directflight and

10 Avalon Aero for their guidance and logistical support during the flight campaign. We also acknowledge the contribution of the UK National Environmental Research Council (NERC), the Ministry of Earth Sciences, Government of India and the principal investigators of the Monsoon program both in the UK and India. Daniel Say was funded by a NERC studentship and grant NE/M014851/1. Anita Ganesan was supported by a NERC Independent Research Fellowship NE/L010992/1. Mark Lunt was supported by NERC grants NE/I027282/1 and NE/M014851/1. Funding for the field campaign was made possible by NERC grant NE/I027282/1.





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





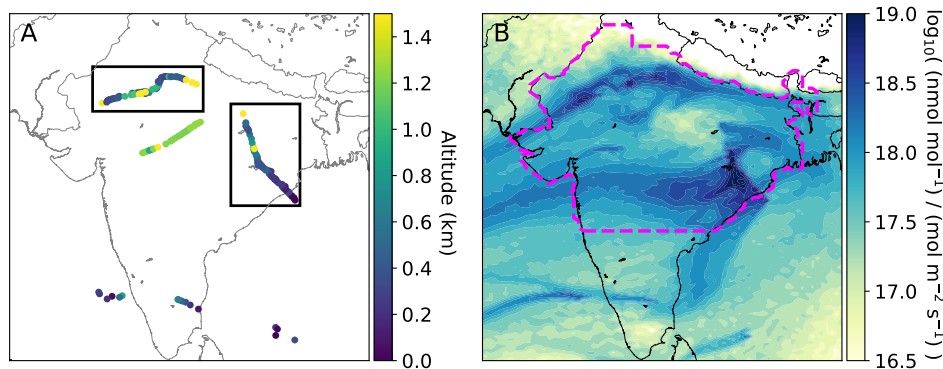

**Figure 1.** (A) The location of aircraft samples collected over India, plotted with respect to altitude. Flight paths represented by the samples outlined in black boxes were repeated three times across the sampling period. (B) Average sensitivity of the flask samples collected over India to emissions originating from the surface. The region denoted as northern-central India is shown in the dashed magenta outline. Reproduced from our companion paper, Say et al., 2018.





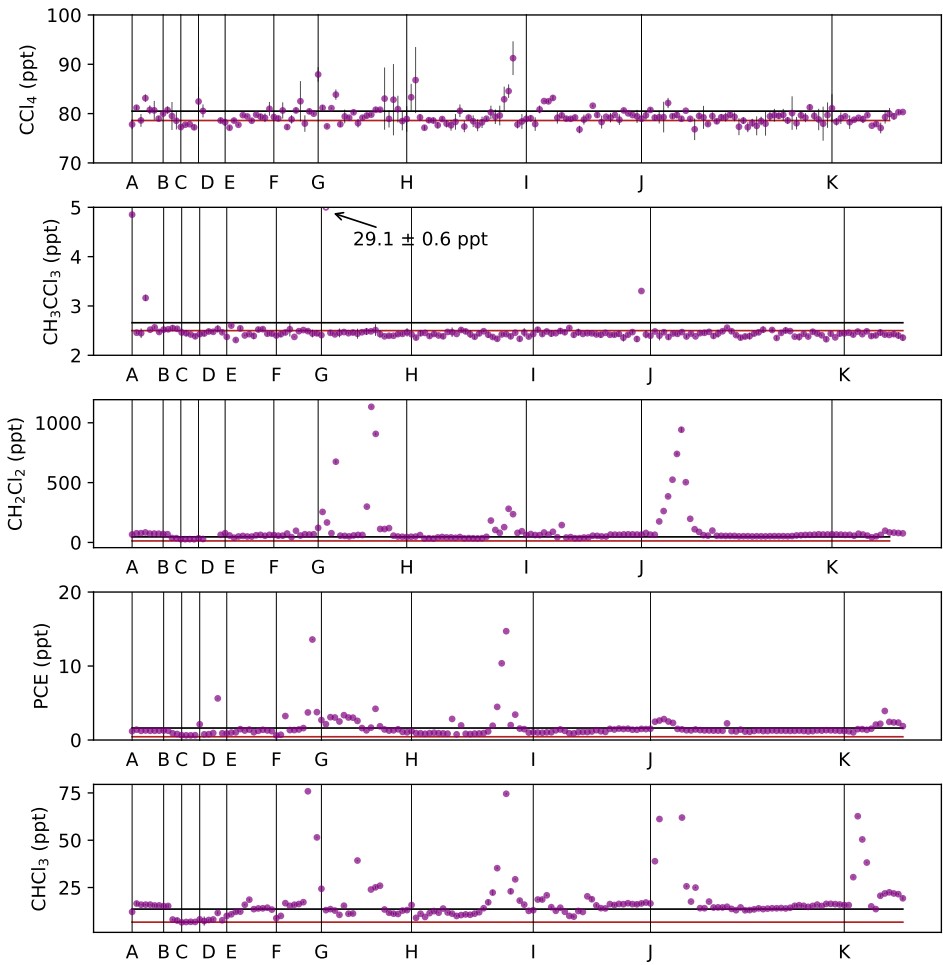

**Figure 2.** Chlorocarbon mole fraction data from 176 flask samples collected over India. Instrumental precision is defined as the standard deviation of the bracketing reference standards for each measurement. Statistical baselines, derived from observations at Mace Head, Ireland and Cape Grim, Tasmania, are shown as black and red lines respectively. Observations are split by flight (black vertical dividers), with letters A-K representing the flights detailed in Table 2.



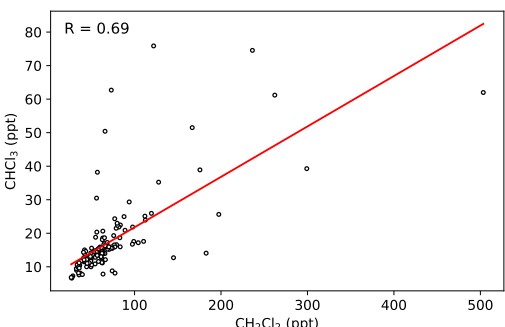

**Figure 3.** Scatter plot of dichloromethane (DCM) and chloroform mole fraction data with Pearson correlation coefficient.





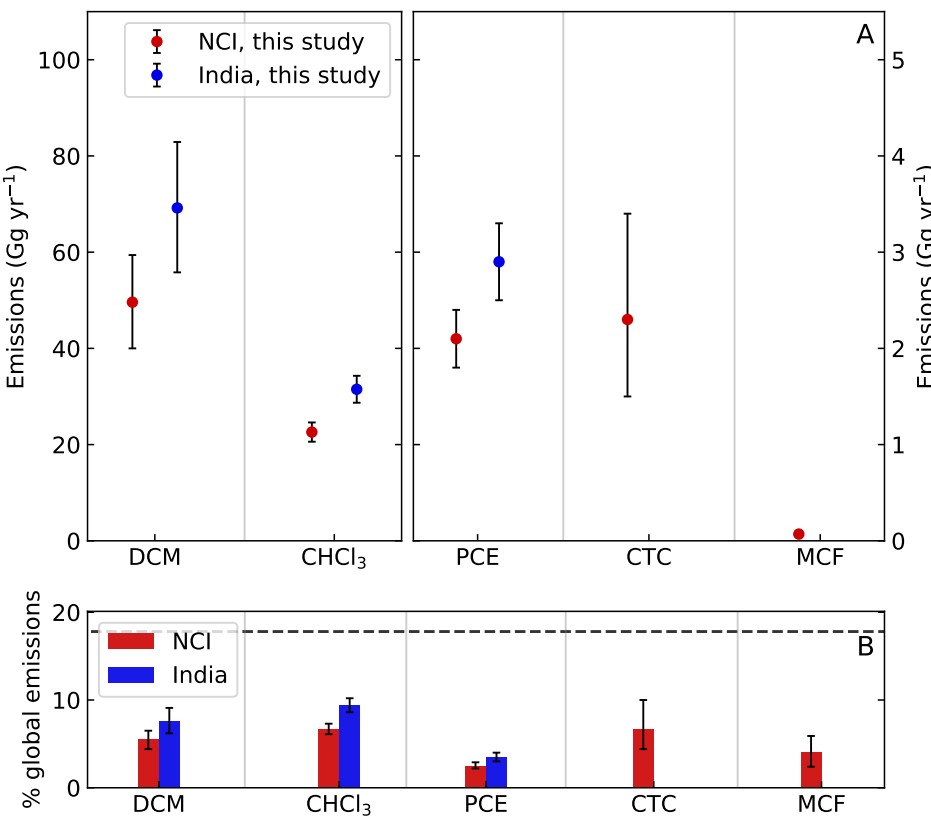

**Figure 4.** NCI (red) and India total (blue) chlorocarbon emissions (Gg yr$^{-1}$) derived in this study. (B) The estimated contribution of the NCI and India to global chlorocarbon emissions (global estimates were derived using the AGAGE 12-box model and are an extension of the work by Rigby et al. (2014)). The dashed line represents India's percentage of the global population in 2016. Error bars represent the $5^{th}$ - $95^{th}$ percentiles of the posterior distribution. Since the remaining sources of CTC and MCF are highly uncertain and are not necessarily linked to population density, emissions of these gases were not scaled to estimate a national total.





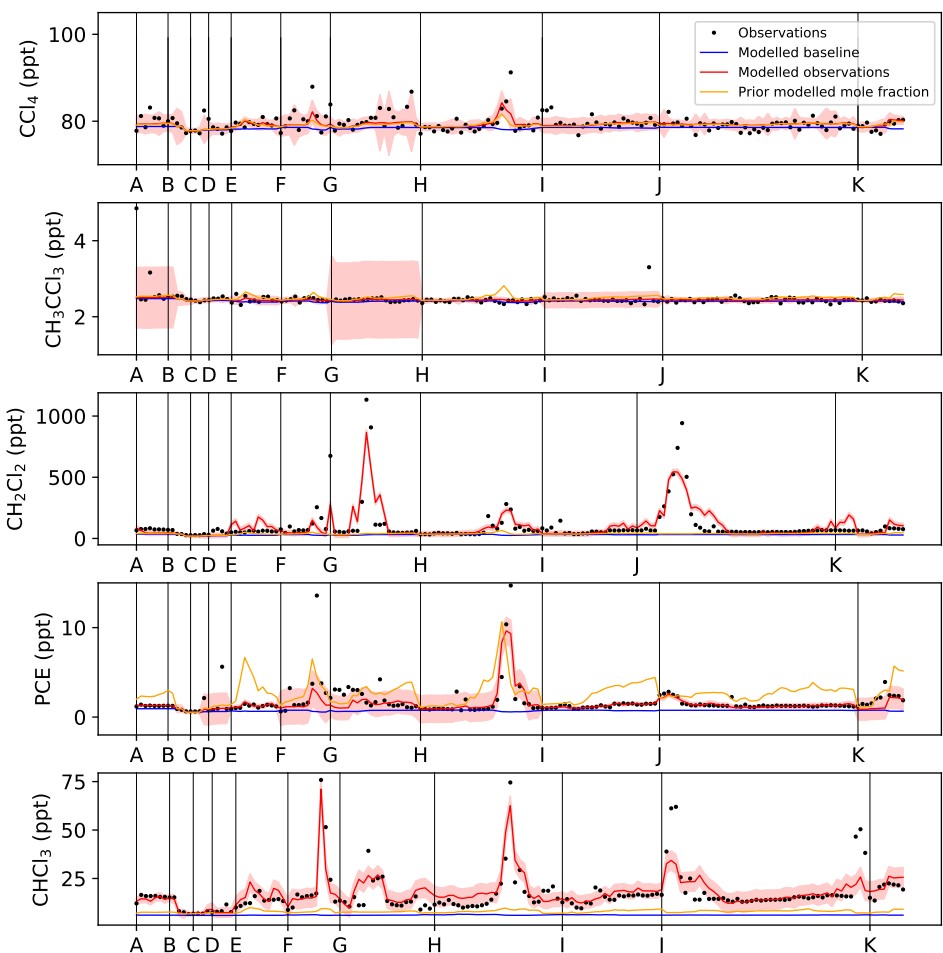

**Figure 5.** Comparison of measured (black points) with prior (orange line) and posterior (red line) mole fractions. The shading represents the model-measurement uncertainty. For all gases, prior emissions were distributed according to the NOAA night light distribution. Each time-series is divided by flight (black vertical dividers), with letters A-K representing the flights detailed in Table 2.





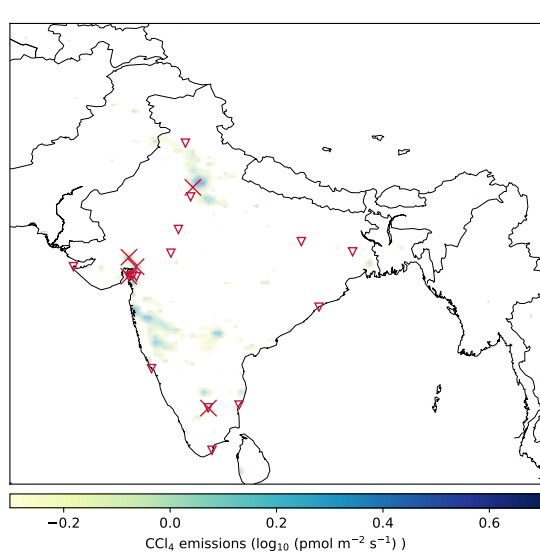

**Figure 6.** Posterior CTC emissions distribution. The known locations of chloromethane production facilities (crosses) and chlor-alkali plants (open triangles) are also shown.



**Table 1.** Summary of chlorocarbon atmospheric lifetimes, ozone-depletion potentials (ODP) and applications. Unless stated, values are from the 2014 Scientific Assessment of Ozone Depletion (Carpenter et al., 2014). *Parameters derived for 30 °N - 60 °N.

| Chlorocarbon | Atmospheric lifetime | ODP | Applications |
|---|---|---|---|
| Carbon tetrachloride, CTC | $26 \pm 17\%$ years | 0.72 | Cleaning agent |
| Methyl chloroform, MCF | $5.0 \pm 3.0\%$ years | 0.14 | Cleaning agent, degreaser |
| Dichloromethane, DCM | 144 days (Feng et al., 2018) | Not well quantified | General purpose solvent, feedstock, foam blowing agent |
| Chloroform | 149 days (Montzka et al., 2011) | Not well quantified | Feedstock |
| Perchloroethene, PCE | 111 days* | 0.005* | Dry cleaning agent |



**Table 2.** India flight campaign information. IST – India Standard Time, N – Number of samples. Flight labels correspond to the dividers shown in Figs. 2 and 5. Table is also shown in our companion paper, Say et al., 2018.

| Flight number (flight label) | Date (time, IST) | Sampling region | Mean altitude (range, km) | Number of samples |
| --- | --- | --- | --- | --- |
| B957 (A) | 12/06 (06:02 – 07:55) | NE India | 1.20 (0.30 – 7.40) | 9 |
| B959 (B) | 21/06 (08:10 – 08:21) | S India | 0.46 (0.05 – 0.87) | 2 |
| B963 (C) | 25/06 (16:52 – 18:00) | S India | 0.31 (0.21 – 0.53) | 4 |
| B966 (D) | 27/06 (07:12 – 09:49) | S India | 0.30 (0.02 – 0.66) | 9 |
| B968 (E) | 30/06 (05:03 – 06:51) | NW India | 0.98 (0.28 – 3.15) | 11 |
| B969 (F) | 02/07 (05:21 – 07:11) | NW India | 0.53 (0.28 – 0.64) | 11 |
| B971 (G) | 04/07 (07:23 – 08:57) | NE India | 0.38 (0.02 – 1.65) | 20 |
| B972 (H) | 05/07 (05:23 – 07:06) | NW India | 0.83 (0.30 – 1.65) | 27 |
| B974 (I) | 07/07 (06:22 – 07:30) | NW India | 1.29 (0.88 – 2.90) | 26 |
| B975 (J) | 09/07 (06:31 – 08:14) | NE India | 0.37 (0.02 – 1.16) | 44 |
| B976 (K) | 10/07 (06:37 – 07:32) | NW India | 0.42 (0.35 – 0.53) | 12 |





**Table 3.** Summary of emissions derived for the NCI and India in Gg yr$^{-1}$, and the contribution of India to global emissions estimated using the AGAGE 12-box model. For each gas, uncertainties correspond to the $5^{th}$ and $95^{th}$ percentiles of the posterior distribution. *Estimated contribution to global emissions from the NCI.

| Chlorocarbon | Prior NCI | Posterior NCI | Posterior India | % Global |
|---|---|---|---|---|
| CTC | 2.0 | 2.3 (1.5 - 3.4) | - | 6.8 (4.4 – 10.0)* |
| MCF | 0.2 | 0.07 (0.04 - 0.10) | - | 4.1 (2.4 – 5.9)* |
| DCM | 14.6 | 49.6 (40.0 - 59.4) | 69.2 (55.8 - 82.9) | 7.6 (6.2 - 9.1) |
| PCE | 4.3 | 2.1 (1.8 - 2.4) | 2.9 (2.5 - 3.3) | 3.5 (3.0 - 4.1) |
| Chloroform | 2.2 | 22.6 (20.6 – 24.6) | 31.5 (28.7 – 34.3) | 9.4 (8.6 – 10.2) |





**Table 4.** Comparison of NCI posterior emission estimates; 1) derived using a night-light distributed prior, 2) derived using a spatially uniform prior, 3) derived using a night-light distributed (CTC, MCF, DCM, PCE) or spatially uniform (chloroform) prior and a filtered dataset, whereby observations corresponding to times at which the NAME simulated and observed wind speed/wind direction differed by more than 20% were removed; in Gg yr$^{-1}$. For each gas, uncertainties correspond to the $5^{th}$ and $95^{th}$ percentiles of the posterior distribution.

| Chlorocarbon | 1) Posterior NCI – night-lights | 2) Posterior NCI – spatially uniform | 3) Posterior NCI – filtered met. |
|---|---|---|---|
| CTC | 2.3 (1.5 - 3.4) | 2.0 (1.4 – 2.9) | 2.5 (1.7 – 3.7) |
| MCF | 0.07 (0.04 - 0.10) | 0.07 (0.04 – 0.10) | 0.08 (0.05 – 0.11) |
| DCM | 49.6 (40.0 - 59.4) | 49.3 (44.2 – 54.5) | 51.1 (42.6 – 61.1) |
| PCE | 2.1 (1.8 - 2.4) | 2.2 (1.8 – 2.5) | 2.2 (1.8 – 2.7) |
| Chloroform | - | 22.6 (20.6 – 24.6) | 21.9 (19.4 – 24.5) |