# Peer review of "Atmospheric observations and emission estimates of ozone-depleting chlorocarbons from India"

_Atmospheric Chemistry and Physics, 2018_

## Referee Comment (RC1) · Anonymous Referee #1 · 22 Feb 2019

Review of Atmospheric observations and emission estimates of ozone-depleting chlorocarbons from India, submitted to ACPD

General remarks: This is a well-written sound study on Indian emissions of long-lived and short-lived chlorocarbons from India. This has been a notoriously undersampled region of the world so far and therefore even if there is only 1 month of measurements available, this should be published. I therefore suggest publishing the manuscript in ACP, taking into account the suggestions from below.

P 1 Line 8: This has only been 1 month of measurements not 2.

P 2 Line 8: There have been updates to this numbers in Carpenter et al. (2014) and in

[Figure]

Liang et al. 2018.

P 2 Line 17: ODPs

P 2 Line 19: What about the new Chapter 1 of the Ozone Assessment (Engel and Rigby, 2019)

P 2 Line 21 and 22: Hossaini et al and Fang et al is plural therefore, show and estimate

P5 L25ff. Somehow it is unusual to use different a priori estimates for the individual compounds. especially questionable in this respect is the use of top-down estimates as an a priori which should be independent of top-down estimates. I suggest that you use the AGAGE-12-box based method for all compounds.

P9 L13 The focus on chloro-alkali plants is a misinterpretation of the literature. It is the total of the production of chlorine related products (chloro-alkane production and chloro-alkali plants). Citation from the conclusion of Hu et al.; Our findings suggest that the majority of US CCl4 emissions could be related to industrial sources associated with chlorine production and processing

P9 L16ff What about the correlation of CCl4 with CHCl3. If there is co-production with CH2Cl2, there should also be co-production with CHCl3, please discuss.

P11. L13 . . .long-lived chlorocarbons. . .

P22. Table 2. The new Ozone Assessment has the lifetime of CCl4 as 32 years. Please correct and cite accordingly.

---

## Referee Comment (RC2) · Anonymous Referee #2 · 22 Mar 2019

My main comment in the quick report was: What sets this manuscript apart from its companion paper (acp-2018-1146, Emissions of CFCs, HCFCs and HFCs from India). Both report synthetic halocarbon measurements from the same campaign which are even shown to partly correlate with each other due to similar sources. The authors responded as follows:

"Our companion paper 'Emissions of CFCs, HCFCs and HFCs from India [revised to 'Emissions of CFCs, HCFCs and HFCs from India based on atmospheric measurements']' focuses on a suite of gases that are used extensively as refrigerants and foam blowing agents, whereas the primary application of the chlorocarbons discussed in this

paper are as solvents. While the majority of gases mentioned in the companion paper are emitted predominantly from residential and mobile sources, chlorocarbons are typically associated with industrial sources such as manufacturing facilities. In addition, the analysis of chloroform in particular requires consideration of biogenic sources not shared by any of the CFCs, HCFCs or HFCs. While the companion paper is framed in the context of the Paris Agreement (emissions totals quoted using global warming potentials in carbon dioxide equivalents), this manuscript focuses exclusively on the potential significance of Indian chlorocarbons emissions as threats to the recovery of the ozone layer (emissions totals quoted in ODP Gg yr-1, where possible). Hence, we feel justified in presenting the Monsoon measurements as two separate manuscripts. However, we do agree that there is the need to frequently look up details from the companion, and that this is a limitation of the current manuscript. To address this issue, we add complete experimental details to the Materials and Methods section, including a full description of the flask sampling routine and analysis, simulation of back-trajectories using NAME, assignment of boundary conditions and estimation of global emissions using the AGAGE 12-box model."

This justification is flawed. As stated in the companion paper, CFC-113 is a solvent. Various publications have shown that emissions of HCFC-22 and HFC-23 are dominant in distinct industrial areas, see, e.g., Fang et al., EST, 2015. Several of the gases in the companion paper are therefore not emitted predominantly from residential and mobile sources. The authors themselves link DCM and chloroform to HFC-32 and HCFC-22 through chemical manufacturing processes as confirmed by their own atmospheric correlations. So the species in the two papers are heavily linked and in my opinion it still does not make sense to discuss these links in each of the companion papers. In addition, half of the species reported in the companion paper are ODSs, so their emissions are a threat to the recovery of the ozone layer, too. Their potential for that is however not even mentioned in this manuscript. The duplication of the experimental details in the Materials and Methods section only adds to these concerns. I therefore still think that the authors should revisit where to draw the lines between the two papers,

or whether there is actually a need to split this good work into two half-duplicates.
* * *

---

## Referee Comment (RC3) · Anonymous Referee #1 · 4 Apr 2019

After reading the comment of reviewer #2, I have to admit that my reviewer #2 made a really good comment. He/she expressed my underlying feeling in a very appropriate manner. The separation into 2 papers does not make sense. In addition, as already mentioned by reviewer #2 some sources could be identical, such as HCFC-22 and CHCl3. Therefore, I would also recommend that the 2 papers should be combined in order to create an additional value for the reader.